# Novel Core Gene Signature Associated with Inflammation-to-Metaplasia Transition in Influenza A Virus-Infected Lungs

**DOI:** 10.3390/ijms252211958

**Published:** 2024-11-07

**Authors:** Innokenty A. Savin, Aleksandra V. Sen’kova, Elena P. Goncharova, Marina A. Zenkova, Andrey V. Markov

**Affiliations:** Institute of Chemical Biology and Fundamental Medicine, Siberian Branch of the Russian Academy of Sciences, Lavrent’ev Avenue 8, 630090 Novosibirsk, Russia; savin_ia@niboch.nsc.ru (I.A.S.); egn@niboch.nsc.ru (E.P.G.); marzen@niboch.nsc.ru (M.A.Z.); andmrkv@gmail.com (A.V.M.)

**Keywords:** influenza A virus, SARS-CoV-2, inflammation, metaplasia, regeneration, bioinformatics, in vivo, lung neoplasms

## Abstract

Respiratory infections caused by RNA viruses are a major contributor to respiratory disease due to their ability to cause annual epidemics with profound public health implications. Influenza A virus (IAV) infection can affect a variety of host signaling pathways that initiate tissue regeneration with hyperplastic and/or dysplastic changes in the lungs. Although these changes are involved in lung recovery after IAV infection, in some cases, they can lead to serious respiratory failure. Despite being ubiquitously observed, there are limited data on the regulation of long-term recovery from IAV infection leading to normal or dysplastic repair represented by inflammation-to-metaplasia transition in mice or humans. To address this knowledge gap, we used integrative bioinformatics analysis with further verification in vivo to elucidate the dynamic molecular changes in IAV-infected murine lung tissue and identified the core genes (*Birc5*, *Cdca3*, *Plk1*, *Tpx2*, *Prc1. Rrm2*, *Nusap1*, *Spag5*, *Top2a*, *Mcm5*) and transcription factors (*E2F1*, *E2F4*, *NF-YA*, *NF-YB*, *NF-YC*) involved in persistent lung injury and regeneration processes, which may serve as gene signatures reflecting the long-term effects of IAV proliferation on the lung. Further analysis of the identified core genes revealed their involvement not only in IAV infection but also in COVID-19 and lung neoplasm development, suggesting their potential role as biomarkers of severe lung disease and its complications represented by abnormal epithelial proliferation and oncotransformation.

## 1. Introduction

Influenza A virus (IAV) infection causes acute respiratory illness with potentially serious and fatal complications [1]. IAV can directly infect and destroy lung epithelial cells and alveolar macrophages to induce an immune response and lung injury [2]. Therefore, viral pathogenesis involves not only the direct cytopathic effect of IAV but also an exacerbated host inflammatory response. The airway epithelium is now recognized as a central point in the orchestration of pulmonary inflammation and immunity. It acts as the first line of defense against pathogens and is deeply involved in the regulation of both innate and adaptive immune responses to these challenges [3].

Acute IAV infection (lasting 10 or fewer days) involves widespread epithelial damage with multifocal destruction and desquamation of the tracheal, bronchial, and alveolar epithelium, resulting in large areas virtually devoid of alveolar epithelial cells [4,5]. Airspace edema, hemorrhages, hypercellularity of the alveolar walls and flooding of the alveolar lumina with inflammatory cells are also present [4,5]. In later stages of the disease (duration of more than 10 days), regeneration of the bronchiolar and alveolar epithelium is evident given the necessity to reconstitute effective barriers over such a large area [6]. The alveolar epithelium is remarkably regenerative, and this process, which is normally quite slow and stimulated after injury, can occur with a shift toward either normal or dysplastic repair.

Alveolar epithelial type II cells (AEC2s), together with undifferentiated distal airway stem/progenitor cells, are the major contributors to alveolar maintenance and repair [7,8]. In the injured lung parenchyma, the remaining AEC2s and progenitors proliferate and migrate intensively to occupy the most damaged sites depleted of mature lineages, whereupon they differentiate toward mature epithelium, promote recovery, and give rise to alveolar epithelial type I cells (AEC1s) reconstituting normal alveolar epithelial barriers [8,9]. However, dysregulated proliferation of the epithelium, which progressively spreads from the terminal bronchioles to the alveoli during the recovery phase, is generally accompanied by squamous metaplasia and AEC2 hyperplasia, resulting in the excessive formation of adenomatous structures, which in turn leads to alveolar–capillary blood–gas exchange dysfunction and hypoxemic respiratory failure, which has an extremely negative impact on the general status, especially in immunocompromised patients [5,7,10]. Epithelial hyperplasia was documented as long as 90 days after infection, suggesting a potentially long-lasting impact of IAV on the lungs [11].

Since influenza infection is a multifaceted disease, involving multiple biological pathways [12,13], there is still uncertainty regarding the genes that regulate the development and progression of the disease, its outcomes, and long-term consequences. Recently, Bibert et al. [14] and Kulasinghe et al. [15] performed transcriptomic analysis of the gene expression profiles of peripheral blood and lung samples obtained from coronavirus disease 2019 (COVID-19)- and influenza virus-infected patients but did not delve deeper into the biological machinery that regulates the development of these diseases. Liu et al. [16] performed a deeper analysis of the transcriptomic profiles of peripheral blood from IAV-infected patients; however, the analysis was focused on the immune system-associated cells only and did not touch upon the lung tissue, where IAV-induced hyperplastic processes occur.

In our research, we sought to address the existing knowledge gap regarding the processes of development and normal/dysplastic repair in influenza-infected lungs. We estimated the dynamics of inflammation and metaplastic changes in the lung tissue of IAV-challenged mice during infection development, performed a re-analysis of available transcriptomic data relevant to the processes under investigation, and identified key differentially expressed genes (DEGs) and transcription factors (TFs) associated with the development of IAV infection and the IAV-induced inflammation-to-metaplasia transition process with further verification on the material obtained from IAV-infected mice. Furthermore, the potential involvement of the identified DEGs and TFs in the development of COVID-19 and lung neoplasms was evaluated by a re-analysis of transcriptomic data obtained from human patient material.

## 2. Results

### 2.1. Viral Titers and Morphological Changes in the Lung Tissue of Mice During IAV Infection and Recovery

In the initial phase of this study, we sought to ascertain the temporal parameters associated with the acute response to viral infection in the lung tissue of mice, as well as the formation of hyperplastic changes. To address this issue, Balb/C mice were intranasally (i.n.) infected with a single 50% lethal dose (1 LD_50_) of IAV, a viral dose that typically results in the development of characteristic changes in the lungs [17]. On days 1, 3, 7, 10 and 14 post infection (d.p.i.), lung viral titers were evaluated, and histological examination of the lungs was performed (Figure 1A).

A detectable viral titer was found in only one mouse out of five studied at 1 d.p.i. (5 × 10^2^ focus-forming units (FFU)/(mL × g)), while persistence of the IAV infection to 3 d.p.i. resulted in a dramatic increase in the viral titers in the lungs of mice to a value of 7.9 × 10^4^ FFU/(mL × g). A comparable level of viral titers was maintained in IAV-infected mice up to 7 d.p.i. (6.3 × 10^4^ FFU/(mL × g)), after which (10 d.p.i.), it remained in only one mouse at 10 FFU/(mL × g). At 14 d.p.i., it became undetectable in all mice studied (Figure 1B).

The histological examination revealed changes in the lung tissue of IAV-infected mice that correlated well with the viral load observed and was represented by two main patterns, including parenchymal lung inflammation, apparent during active viral replication (early stage), and alveolar epithelial metaplasia, clearly evident after the clearance of virus and resolution of infection (later stage) (Figure 1C).

The pathological changes in the lung tissue at the early stage of IAV infection were evidenced by neutrophil-dominated inflammatory infiltration with interstitial and alveolar edema and hemorrhage, which was most pronounced at 3 and 7 d.p.i. and accompanied by high viral titers in the lungs (Figure 1B–D). In the later stage of IAV infection, the intensity of inflammation in the lung tissue was reduced by 2.3- and 5.4-fold at 10 and 14 d.p.i., respectively, compared to the maximum values at 7 d.p.i., which was accompanied by a drop in viral titers for the corresponding time points (Figure 1B–D). Cellular infiltration was represented mainly by lymphocytes and macrophages with a reduction in the neutrophil content. Foci of epithelial proliferation extending from the terminal bronchioles to the alveoli started to appear from 10 d.p.i., indicating the launch of inflammation-to-metaplasia transition (Figure 1C,D). At 14 d.p.i., proliferation and squamous metaplasia of the alveolar epithelium was fully developed and increased by 4.1 times compared to at 10 d.p.i., occupying a significant part of the lung parenchyma, while the virus was completely eliminated (Figure 1B–D). It appears that precisely within the 7–10-day window following IAV infection, the lung tissue undergoes a transition from inflammation to metaplasia (inflammation-to-metaplasia transition). This transition represents a critical juncture in the course of IAV infection, whereby a decision regarding the consequences of the infection is made: either normal resolution of acute inflammation through normal cell repair, or alternatively, its progression to a chronic form accompanied by dysplastic repair and metaplasia development.

### 2.2. Core Differentially Expressed Genes (DEGs) Related to Acute Inflammation-to-Metaplasia Transition

Based on the histological findings (Figure 1C,D), we further focused our attention on the transitory period of IAV infection (7–12 d.p.i.), when the inflammation-related changes in IAV-infected lung tissue are replaced by metaplasia-associated processes. Considering this period, two independent batches of cDNA microarray datasets obtained from lung tissue of IAV-infected mice were selected. The first batch, related to 7 d.p.i., included GSE43302, GSE56433, and GSE64798. The second batch, related to 10 and 12 d.p.i., included GSE42638, GSE57452, and GSE67241. It should be noted that the microarray datasets were selected independently of the IAV strains used in order to obtain data that are more accurate and virus strain independent. Moreover, datasets obtained from pandemic IAV strains were excluded from the analysis, as they induce substantially more pronounced, even explosive, and destructive changes in lung tissue compared to non-pandemic strains.

The analysis of DEGs using the GEO2R tool (https://www.ncbi.nlm.nih.gov/geo/geo2r/, accessed on 4 November 2024) (Log_2_ (fold change) ≥ 1.5, *p* ≤ 0.05) and Venn diagram reconstruction identified 729 and 42 common DEGs in the first (7 d.p.i.) and second (10–12 d.p.i.) batches, respectively (Figure 2A, left and right diagrams). The observed significant reduction in the number of DEGs during the development of IAV infection in the lung tissue was anticipated, given the declining intensity of the inflammatory process in the lungs (Figure 1D) and the onset of a transition from inflammation to regeneration.

Given that the number of common DEGs identified at 10–12 d.p.i. was significantly lower (42) than at 7 d.p.i. (729), the inclusion criteria for batch #2 were extended to include those genes that were differentially expressed in at least two of the three GEO datasets analyzed, as in the previously published work [18]. This resulted in the number of DEGs considered in batch #2 being increased to 294 (Figure 2A, right diagram, genes circled in red). It is noteworthy that a considerable proportion of the selected DEGs in batch #2 (118 out of 294, 40.1%) were involved in the development of IAV infection at an earlier time point of 7 d.p.i. (Figure 2A, middle diagram). This observation suggests the potential existence of a subset of consistently expressed genes in the tissue of IAV-infected lungs during the transitional period from acute inflammatory response to metaplastic changes.

### 2.3. Identification of Gene Clusters Associated with Inflammation-to-Metaplasia Transition in IAV-Infected Lungs

Next, in order to analyze the interconnectedness of common DEGs identified at 7 and 10–12 d.p.i., corresponding gene association networks were reconstructed from these gene sets using the Search Tool for the Retrieval of Interacting Genes/Proteins (STRING) database. As shown in Figure 2B, DEGs from both explored batches formed tightly interconnected gene networks, characterized by similar structure, despite the significantly different number of uploaded nodes. Additional topologic analysis of the reconstructed gene networks has further confirmed their similarities, since both networks contained the same hub genes with the highest degree centrality score, namely, regulators of cell cycle (*Ccnb1*, *Cdc20*, and *Cdca8*), DNA transcription and replication (*Top2a*), and cell survival (*Birc5*). Interestingly, despite the more than two-fold difference in the number of nodes (Figure 2B) in the networks related to batch #1 (729 DEGs) (Figure 2B, left network) and batch #2 (294 DEGs) (Figure 2B, right network), their MCODE analysis revealed gene clusters with comparable sizes of 42 and 35 nodes, respectively (Figure 2C, left and right diagrams), with around 40% of nodes being common for both analyzed batches (Figure 2C, middle diagram).

The performed gene network analysis corroborated our hypothesis regarding the presence of core genes that are consistently expressed in lung tissue throughout the progression of IAV infection and are associated with the transition from an acute inflammatory response (batch #1) to metaplastic and regenerative changes (batch #2), which includes key regulators of cell proliferation.

### 2.4. Analysis of Core Genes with Similar Expression Profiles Using STEM Clustering Algorithm

Given the evidence that a subset of highly interconnected genes in gene networks may demonstrate a similar pattern of expression [19], we have further investigated whether the revealed genes from the MCODE clusters demonstrate comparable expression profiles. To address this issue, the most representative GEO datasets, GSE43302 and GSE67241, from batches #1 and #2, respectively, were clustered in a time-dependent manner on gene expression using the short time expression miner (STEM) tool, which is specifically designed for the analysis of short time-series gene expression datasets [20]. The clustering analysis revealed a high degree of similarity in the expression patterns of the DEGs from the analyzed MCODE groups. Specifically, 23 out of 42 genes from batch #1 and 33 out of 35 genes from batch #2 showed close expression patterns with gradual up-regulation during IAV infection (Figure 3A,B). A further functional analysis of these genes revealed their association with the regulation of cell cycle, including mitotic spindle assembly, regulation of sister chromatid segregation, and microtubule cytoskeleton organization, as well as cell division in general, including cell cycle checkpoints and nuclear division (Figure 3C,D).

Given that the primary objective of this study was to identify the core genes associated with the inflammation-to-metaplasia transition and activated in lung tissue in response to IAV infection and IAV-induced earlier metaplastic changes, we have finally selected a subset of genes common to both STEM-clustered groups at 7 d.p.i. and 10–12 d.p.i., namely, *Birc5*, *Cdca3*, *Mcm5*, *Nusap1*, *Plk1*, *Prc1*, *Rrm2*, *Spag5*, *Top2a*, and *Tpx2* (Figure 3E). The expression levels of this set of genes were further validated in the animal model.

### 2.5. Identification of Potential Transcription Factors Regulating the Expression of Core DEGs Related to Inflammation-to-Metaplasia Transition During IAV Infection Development

Considering the fact that similar expression patterns of DEGs may indicate the presence of common master regulators [19], *cis*-regulatory elements within the promoters of genes selected through STEM clustering (Figure 3A,B) were further examined using the iRegulon plugin in Cytoscape software. The promoter analysis indicated that E2f4 is the most probable transcription factor (TF) regulating the expression of the DEGs from both batches, with a Normalized Enrichment Score (NES) of 13.3 (Figure 3F). In addition to E2f4, another member of the E2F transcription factor family, E2f1, was also identified as a specific regulator of the expression levels of the DEGs from batch #1. This fact agrees well with the known involvement of the E2F protein family in the regulation of the cell cycle-related processes mentioned above (Figure 3C) [21,22,23,24]. In the case of the early metaplastic stage (10–12 d.p.i.), the cell cycle-associated transcription factor Nfya (NES = 11.0) was additionally revealed as a probable upstream regulator of batch #2-related DEGs (Figure 3F).

Based on the obtained promoter analysis data, E2f1, E2f4, Nfya, as well as Nfyb and Nfyc, due to the trimeric structure of NF-Y transcription factors [25], along with their targeted DEGs, were chosen for further validation.

### 2.6. Analysis of DEG and TF Expression Patterns Associated with IAV Infection Development

The expression levels of core DEGs (*Birc5*, *Cdca3*, *Plk1*, *Tpx2*, *Prc1*, *Rrm2*, *Nusap1*, *Spag5*, *Top2a*, and *Mcm5*) and TFs (*Nfya*, *Nfyb*, *Nfyc*, *E2f1*, and *E2f4*) were evaluated in the lung tissue of IAV-infected mice at 1, 3, 7, 10, and 14 d.p.i. by TaqMan-based qRT-PCR. Additionally, their expression profile changes were visualized through heatmaps, and correlation analysis of the expression patterns of the above-mentioned core genes and TFs was performed. The results are summarized in Figure 4.

The lung tissue of healthy animals was characterized by low expression levels of the studied genes, which were set as 1, while IAV infection caused significant changes in their expression, the profile of which was comparable to the results of the STEM clustering analysis mentioned above (Figure 3A,B). The majority of analyzed genes, including *Birc5*, *Nusap1*, *Plk1*, *Prc1*, *Rrm2*, *Spag5*, *Top2a*, and *Tpx2*, demonstrated an ascending expression pattern consisting of a gradual up-regulation, reaching a local maximum at 7 d.p.i., followed by a slight drop at 10 d.p.i. with subsequent increases in gene expression at 14 d.p.i. to maximal values (Figure 4A). The expression patterns of *Cdca3* and *Mcm5* were also characterized by a time-dependent gradual up-regulation; however, unlike the rest of the DEGs, their short-term drop in expression was observed at earlier time points of 3 and 1 d.p.i., respectively (Figure 4A).

Analysis of the TF expression profiles has shown that *Nfya* has the same expression profile as most DEGs, while *E2f4*, *Nfyb*, and *Nfyc* were up-regulated during the whole experiment without a significant drop in expression levels. Intriguingly, in contrast to other TFs, *E2f1* was significantly down-regulated by an average of 4-fold compared to healthy controls over the whole course of the experiment (Figure 4B).

According to the intensity of DEG expression in response to IAV infection development, the analyzed DEGs can be divided into three distinct groups, including (i) DEGs that are the most susceptible to IAV infection, namely, *Birc5* (18.7-fold change at 14 d.p.i.), *Cdca3* (11.9-fold change at 14 d.p.i.), *Plk1* (8.6-fold change at 14 d.p.i.), and *Tpx2* (6.8-fold change at 14 d.p.i.); (ii) DEGs with a moderate response to virus with less than 3-fold expression changes (*Rrm2*, *Prc1*, *Nusap1*); and (iii) DEGs down-regulated at the early period of IAV infection (1–2 d.p.i.) with a subsequent increase in the expression levels up to the healthy controls (*Top2a*, *Mcm5*, *Spag5*) (Figure 4A).

In the case of TFs, as mentioned above, the *E2f1* expression level was characterized by an abrupt down-regulation in response to IAV infection (0.65–0.08-fold changes compared to healthy controls throughout the experiment), while *E2f4* was up-regulated with the most pronounced response at the end of the acute inflammatory stage, reaching 1.7-fold change compared to control (7 d.p.i.) and remaining at this level during the further transition to the metaplastic stage.

We suppose that such an opposite effect of IAV infection development on the expression of *E2f1* (down-regulation) and *E2f4* (up-regulation), both of which are known activators and suppressors of proliferation-related genes, respectively, as previously published [23], can be due to the compensatory reaction to the IAV-induced excessive cell proliferation in lung tissue observed by histological analysis (Figure 1C,D). The subunits of NF-Y were more susceptible to IAV infection compared to *E2f4*: their maximal expression changes were 3.8-, 3.2-, and 4.2-fold for *Nfya* (7 d.p.i.), *Nfyb* (10 d.p.i.), and *Nfyc* (14 d.p.i), respectively, compared to the control (Figure 4B,C).

Finally, to estimate how similar the expression patterns of identified core DEGs and their predicted TFs are to each other during the course of the experiment, correlation analysis between each DEG and TF of interest was performed according to Pearson’s model (Figure 4D).

As shown in Figure 4D, the expression levels of all identified core DEGs throughout the experiment were very similar, regardless of the intensity of their response to IAV infection, demonstrating either a strong (0.6–0.79) or very strong (0.8–1) positive correlation. These data corroborate the results of the STEM clustering analysis (Figure 3A,B), which allows us to consider this set of DEGs as a distinct gene signature associated with IAV-induced metaplasia of lung tissue. Moreover, the results of the correlation analysis also indicate that E2f4, despite its repressor functions, can be considered the most probable regulator of the identified core DEGs, since its correlation coefficients are in the range of 0.69–0.94 (strong and very strong correlation) (Figure 4D). Compared to *E2f4*, the expression profiles of *Nfya*, *Nfyb*, and *Nfyc* were significantly less similar to the expression patterns of the identified core genes, and *E2f1*, despite its negative response to IAV infection, did not correlate with the expression of core genes at all.

Thus, the observed results confirmed the potential role of the identified core DEGs (*Birc5*, *Cdca3*, *Plk1*, *Tpx2*, *Prc1*, *Rrm2*, *Nusap1*, *Spag5*, *Top2a*, and *Mcm5*) as regulators of lung cell proliferation during IAV infection development and pointed to *E2f4* as the most probable regulator of the IAV-induced inflammation-to-metaplastic transition in the lungs.

## 3. Discussion

Influenza A virus (IAV) is one of the most encountered viruses in human history: in the last 100 years alone, human society has faced four major IAV pandemics [26]. Despite such a long history of IAV global spreads, approved therapeutics for IAV treatment remain limited to neuraminidase inhibitors, while newer drug candidates, such as inhibitors of viral RNA-dependent RNA polymerase and acidic polymerase endonuclease, are still in clinical trials and suffer from uncertain clinical efficacy [27]. The susceptibility of IAV to rapid mutation allows it to evade both pharmaceutical intervention and vaccine-induced immune response, which determines the need to develop not only IAV-targeted compounds but also blockers of the delayed deleterious effects of IAV infection in lung tissue.

As influenza is a pulmonary epithelial infection, IAV primarily targets and replicates within respiratory epithelial cells, particularly alveolar epithelial cells (AECs) [11]. After injury, the epithelium can either fully regenerate in form and function or be repaired by dysplastic scar tissue, eventually forming large zones of the lung filled with tumor-like epithelial cells with squamous metaplasia [11,28]. Due to the low expression of canonical alveolar epithelial proteins, these cells are unable to effectively mediate gas exchange, leading to exacerbated hypoxemia associated with delayed viral clearance and prolonged inflammation [29,30]. In the presence of impaired regulatory mechanisms, IAV-induced excessive epithelial proliferation developing after viral clearance can have a negative impact on patients’ respiratory function, reducing their quality of life [31] or even being fatal [32].

Most studies of IAV have focused on the acute stage, with the extensive characterization of viral pathogenesis and antiviral responses. Only a few long-term follow-up studies have demonstrated the prevalence of chronic pulmonary sequelae after recovery from the acute response to IAV infection, associated with reduced diffusion capacity across the blood–gas barrier and lower health-related quality of life (HRQoL) scores [33], which were more severe in elderly subjects, infants, and patients with underlying medical conditions such as ischemic heart disease, diabetes mellitus, and immunosuppressed state [34,35,36]. Because most IAV infections are not fatal and their symptoms last only a few days, the chronic effects of IAV infection tend to be overlooked, highlighting the need for a more detailed study of the long-term consequences of influenza challenge that persist after viral clearance.

In this work, we used a complex in silico workflow, including the analysis of publicly available cDNA microarray datasets from IAV-infected lung tissue of mice and MCODE analysis of gene association networks with further STEM clustering of gene expression patterns, to identify a set of core genes associated with the IAV-mediated inflammation-to-metaplasia transition. As a result, the following genes were selected as likely regulators of the long-term outcome of IAV infection in mice: *Birc5*, *Cdca3*, *Mcm5*, *Nusap1*, *Plk1*, *Prc1*, *Spag5*, *Top2a*, and *Tpx2*. Subsequent validation on material from IAV-challenged mice allowed us to classify these genes into three groups according to the intensity of their response to IAV infection.

The first group consisted of the genes *Birc5*, *Cdca3*, *Tpx2*, and *Plk1*, which are most susceptible to IAV infection. These genes were characterized by the sequential amplification of expression during viral infection, which reached maximum values at the end of the considered time period (12–14 d.p.i.) according to both transcriptomic analysis (Figure 3A,B) and RT-PCR analysis of samples from the performed animal experiment (Figure 4A). Considering that the mentioned core genes are known regulators of cell proliferation and survival (Figure 3C,D), the detected increase in their expression at 12–14 d.p.i. is consistent with active hyperproliferation of the alveolar epithelium in IAV-infected lung tissue at the metaplasia stage observed by histological analysis (Figure 1B). According to published data, these genes are key regulators of the cell cycle [37,38,39]. The protein products of *Cdca3*, *Plk1*, and *Tpx2* mediate the degradation of G2/M checkpoint-associated WEE kinase [38], regulate the interaction of the mitotic spindle with chromatids at kinetochores [39], and regulate the activity of mitotic kinesins [40], respectively. Survivin, encoded by *Birc5*, is also involved in cell cycle regulation as part of the chromosome passenger complex during cell division and also suppresses apoptosis by directly inhibiting caspase 9 activation [37]. The overexpression of the aforementioned genes in murine lungs during the IAV-induced inflammation-to-metaplasia transition is consistent with published data. Previously, *Cdca3*, *Tpx2*, and *Birc5* were reported to be significantly overexpressed in various metaplastic conditions in patients, namely, gastric intestinal metaplasia [40], lung squamous metaplasia and dysplasia [41], and endometrial hyperplasia [42], respectively. In addition, the siRNA-mediated knockdown of *Plk1*, *Tpx2*, and *Birc5* apparently suppressed the viability of human preneoplastic cells from the upper aerodigestive tract [43] and human bronchial epithelial cells [44,45], and pharmacological blockade of Plk1 by BI2536 significantly inhibited the proliferation of primary murine lung epithelial cells [46], confirming the association of these genes with metaplastic cell hyperproliferation. Interestingly, all of these genes have been linked to viral infections to some extent, including the key role of *Birc5* and *Plk1* in the survival of HIV-1- [47] and hepatitis B [48]-infected cells, respectively, the up-regulation of *Cdca3* during dengue fever [49], and the hub position of *Tpx2* in the COVID-19 regulome [50]. However, to our knowledge, the association of these genes with IAV infection has not yet been published.

The second group of core genes with a moderate response to IAV infection development included *Nusap1*, *Prc1*, and *Rrm2*, which are also related to cell cycle regulators. The weaker response of *Nusap1* to IAV infection compared to the first group of core genes can be explained by its critical role in the regulation of spindle assembly [51] and the ability of its overexpression to induce excessive microtubule binding with subsequent cell cycle arrest [52]. Plycomb repressive complex 1 (PRC1), required for the polarization of parallel microtubules and correct assembly of the contractile ring [53], was found to promote the export of IAV ribonucleoprotein and to thus stimulate IAV replication [54]. This property probably determines the observed short-term down-regulation of *Prc1* in the early stage of IAV infection (Figure 4A), when IAV hijacks already synthesized PRC1 and suppresses its expression at the mRNA level as a compensatory response. Ribonucleotide reductase M2 (RRM2) is also involved in the maintenance of viral infection, playing a key role in the synthesis of 2′-deoxyribonucleoside 5′-diphosphates essential for DNA biosynthesis, replication and repair [55,56]. RRM2 was also upregulated in response to hepatitis C virus infection and may regulate hepatitis C RNA synthesis [57].

The third group of core genes characterized by short-term down-regulation at early stages of IAV infection (1–3 d.p.i.) with subsequent fluctuation of expression to non-infected control levels included *Mcm5*, *Spag5*, and *Top2a* (Figure 4A). Similar to the genes described above, these genes also play important roles in cell proliferation. Minichromosome maintenance component 5 (MCM5) is part of the MCM complex regulating the G0 to G1/S cell cycle phase transition [58]. Sperm-associated antigen 5 (SPAG5) is involved in mitotic spindle formation and chromosome segregation [59], with recent studies revealing that SPAG5 also plays an independent role in the prevention of stress-induced cell apoptosis [60] and in the maintenance and activation of the mitochondrial translational stability regulator CLUH, critical for cell survival, as a part of the SPAG5/KNSTRN complex [61]. Consistent with these data, knockdown of *SPAG5* by siRNA was found to significantly suppress the proliferation and motility of AEC2 (A549 cells) [62]. Topoisomerase 2 alpha (TOP2A) is a key enzyme controlling the topology of DNA during transcription [63], maintaining mitotic chromosome structure that prevents cells from prematurely exiting mitosis [64] and, possibly, protecting mitochondrial DNA during and after its replication from increased negative supercoiling, which is critical for cell survival and proper functioning [65]. Only MCM5 has been reported to be involved in the development of IAV infection by stimulating IAV genome replication through the stabilization of viral RNA-dependent RNA polymerase [66]. The association of SPAG5 and TOP2A with viral infections has only been shown in a series of bioinformatic studies, in which SPAG5 and TOP2A occupied nodal positions in IAV- [67] and SARS-CoV-2 [68]-related regulomes, respectively. Considering the role of SPAG5 and TOP2A in the processes concerning cell proliferation and survival, it is highly likely that these molecules are involved in the survival of IAV-infected cells, although the exact mechanism remains to be elucidated. However, the question remains as to the reason for the decreased expression of *Mcm5*, *Spag5*, and *Top2a* at the earliest stages of IAV infection development (Figure 4A). We suppose it may be related to the ability of IAV to induce G0/G1 cell cycle arrest [69], which is necessary for the increased efficiency of viral replication [70], easier disassembly of the endoplasmic reticulum and Golgi apparatus [71], and avoidance of early apoptosis of infected cells [72], but this hypothesis requires additional studies.

Further promoter analysis of the identified core genes with subsequent correlation analysis of experimental data from animal studies revealed E2F4 as their probable master regulator (Figure 4B,D). Traditionally, E2F4 has been considered a transcriptional repressor whose activity is critical for the engagement and maintenance of cell cycle arrest in the G0/G1 phase [24], which, as mentioned above, is a key event in the development of IAV infection [69]. However, a number of studies have also demonstrated the transcription-inducing capacity of E2F4 [73,74]. The results obtained are consistent with published data. Recently, Bertrams et al. identified E2F4 as a candidate transcription factor controlling IAV infection in human primary lung samples [75], and Guo et al. demonstrated its high association with the pulmonary macrophage response during IAV infection in mice [76].

In addition to E2F4, E2F1 and the subunits of NF-Y (Nfya, Nfyb, Nfyc) were also predicted here as core gene-targeting transcriptional regulators (Figure 3F), and despite the weak correlation of their expression profiles with the expression of core genes (Figure 4D), these TFs are sensitive to the development of IAV infection in lung tissue (Figure 4B), which is in good agreement with previously published data. According to Mayank et al., E2F1 is involved in the regulation of apoptosis of IAV-infected cells induced by IAV nucleoprotein [77], and NF-Y has been found to mediate the host cell response to various viruses, including Herpes simplex virus type 1 [78], Epstein Barr virus [79], and human T leukemia/lymphoma virus 1 [80]. Thus, the data obtained may suggest that the IAV-induced inflammation-to-metaplasia transition in lung tissue may be regulated by transcription factors that play a key role in the development of a wide range of viral diseases.

Given the use of IAV-infected mice as the central model of the current study and the significant similarities in the development of influenza infection in mice and humans [81], we next asked whether the revealed core genes are sensitive to IAV infection in humans. To assess this translational interaction, the expression of the genes of interest was next evaluated in IAV-infected patients using four independent cDNA microarray datasets (Figure 5A).

Despite the different biomaterial sources used for the transcriptomic analysis (lung tissue and peripheral blood cells for mice (Figure 2) and humans (Figure 5), respectively), all identified core genes were significantly upregulated in the majority of analyzed human-related transcriptomic datasets (Figure 5A), confirming, at least in part, the existence of the translational bridge between mice and humans in the context of IAV-susceptible regulators of the inflammation-to-metaplasia transition. Interestingly, COVID-19, which differs from influenza infection by a more pronounced fibroproliferative phenotype with extracellular matrix reorganization and hypercoagulation [82], was also characterized by the up-regulation of the genes of interest in peripheral blood cells (Figure 5A). Further detailed analysis of the expression of core genes in patients with different COVID-19 stages revealed a trend towards a consistent increase in their median expression levels according to COVID-19 severity (Figure 5B). Among the analyzed core genes, only *SPAG5* and *TOP2A* were characterized by a statistically significant up-regulation in critical COVID-19 stage compared to mild COVID-19 stage, which is in good agreement with published data. Previously, both *SPAG5* and *TOP2A* were identified as hub genes associated with accelerated COVID-19 infection in lung cancer patients [59]. Furthermore, a pronounced up-regulation of *SPAG5* was observed in the lungs of patients with an active fibrotic process [83], *TOP2A* was found to occupy a hub position in the severe COVID-19 regulome [84], and its inhibitors were predicted by an artificial intelligence approach as promising candidates for the treatment of critically ill COVID-19 patients [85]. Given the ability of SPAG5 to control mTOR signaling by inhibiting mTOR–Raptor association and recruiting Raptor to stress granules [86], the overexpression of SPAG5 in severe COVID-19 may be a cytoprotective response that protects lung cells from death induced by SARS-CoV-2-mediated mTORC1 hyperactivation [87]. On the other hand, SPAG5 was found to be positively correlated with the immune cell infiltration of neoplastic lung tissue [88], suggesting its possible involvement in the regulation of the COVID-19-associated cytokine storm and subsequent lung injury. In the case of TOP2A, Afowowe et al. reported the involvement of TOP2A in RNA virus infection via controlling the replication and transcription of viral RNA genomes [89]. In addition, using connectivity map analysis, Sanchez-Burgos and colleagues recently demonstrated the high similarity in transcriptional signatures of TOP2A inhibitors and SARS-CoV-2-induced hypercytokinemia [90], which, as in the case of SPAG5, may suggest a link between TOP2A and the cytokine storm accompanying critical COVID-19.

Thus, the results obtained may indicate the association of the identified core gene signature not only with the IAV-induced inflammation-to-metaplasia transition but also with the progression of pulmonary epithelial infections in general and, secondly, taking into account the up-regulation of core genes not only in lung tissue but also in peripheral blood cells (Figure 5), may suggest the possibility of further consideration of these genes as potential diagnostic markers for the severity of IAV infection. These suggestions require further detailed studies.

Considering the demonstrated susceptibility of the core genes not only to IAV (Figure 2, Figure 3 and Figure 4) but also to SARS-CoV-2 infection (Figure 5), we next questioned with which human diseases in general these genes might be associated. Analysis of the curated data from the DisGeNET database revealed the association of the genes of interest with 64 different diseases, the majority of which (56 out of 64) were related to neoplasia (Figure 6A), as expected due to the involvement of core genes in the regulation of the cell cycle and cell proliferation (Figure 3C).

Interestingly, half of the genes analyzed, namely, *BIRC5*, *TOP2A*, *PLK1*, and *RRM2*, were associated with lung-specific neoplasms (Figure 6A). In addition, *RRM2* and *PRC1* were associated with some non-tumor diseases such as sclerocystic ovaries and immunoglobulin A glomerulonephritis, whereas *SPAG5* was exclusively associated with pregnancy-related diseases (Figure 6A). Further survival analysis of patients with lung adenocarcinoma from The Cancer Genome Atlas (TCGA) cohort independently confirmed the association of the core genes with lung cancer progression: high expression of all genes of interest showed a significant correlation with low overall survival (Figure 6B). Thus, the obtained results, supported by data from available disease-related transcriptomic databases, clearly demonstrate that *Birc5*, *Nusap1*, *Plk1*, *Prc1*, *Rrm2*, *Spag5*, *Top2a*, and *Tpx2*, identified here as core genes of inflammation-to-metaplasia transition in IAV-infected lungs, are also involved in proliferation-related processes in a variety of severe lung diseases, which may indicate a master regulatory role of these genes in the development of IAV-induced severe lung failure, which requires further knockdown studies.

## 4. Materials and Methods

### 4.1. Mice

Balb/C female mice with an average weight of 20–22 g were obtained from the vivarium of the Institute of Chemical Biology and Fundamental Medicine SB RAS (Novosibirsk, Russia). Mice were kept in plastic cages under normal daylight conditions. Water and food were provided ad libitum. All animal procedures were carried out in strict accordance with the recommendations for the proper use and care of laboratory animals (ECC Directive 2010/63/EU). The experimental protocols were approved by the Committee on the Ethics of Animal Experiments at the Institute of Cytology and Genetics SB RAS (ethical approval number 56 from 10 August 2019).

### 4.2. Virus and Cells

Madin–Darby canine kidney (MDCK) cells were obtained from the Russian Cell Culture Collection (Institute of Cytology RAS, St. Petersburg, Russia) and were cultured in Dulbecco’s modified Eagle medium (DMEM; Sigma-Aldrich, St. Louis, MO, USA) supplemented with 10% heat-inactivated fetal bovine serum (FBS; Dia-M, Moscow, Russia) and antibiotic–antimycotic solution (100 U/mL penicillin, 100 μg/mL streptomycin, and 0.25 μg/mL amphotericin; Central Drug House Pvt. Ltd., New Delhi, India). Cells were incubated at 37 °C in a humidified atmosphere containing 5% CO_2_. The influenza A/WSN/33 (H1N1) virus was amplified and titrated in MDCK cells and stored at −80 °C until use. The titer of virus stocks was determined by the focus forming assay (FFA) described below.

### 4.3. Influenza A Virus (IAV) Infection Model

Balb/C mice were anesthetized by isoflurane inhalation containing 3% isoflurane and 97% air with a flow rate 2 L/min and then infected intranasally (i.n.) with a single LD_50_ (the amount of virus that is sufficient to kill 50 percent of mice) of IAV in 40 µL of PBS. The detection of viral titers and histological analyses were carried out on days 1, 3, 7, 10, and 14 post infection (d.p.i.). The lungs of mice were collected and weighed at a specified time. DMEM supplemented with antibiotic–antimycotic solution and 2 μg/mL N-*p*-Tosyl-l-phenylalanine chloromethyl ketone (TPCK)-treated trypsin (infection medium) was added to lungs at a ratio of 1:10 (*v*/*v*) of lung tissue to the medium. Homogenates were prepared using a Sonopuls HD 2070 Ultrasonic Homogenizer (Bandelin, Berlin, Germany). The viral titers were determined using FFA.

### 4.4. Focus Forming Assay (FFA)

Viral titers were determined by the FFA in MDCK cells. Lung homogenates were serially diluted 10-fold with an infection medium and incubated with MDCK cells for 24 h at 37 °C with 5% CO_2_. MDCK monolayers were washed twice with PBS and subsequently fixed with ice-cold 80% acetone for 15 min at room temperature. Viral foci were stained using a mouse monoclonal antibody against influenza A NP (MAB8258, Millipore, Burlington, MA, USA) as described previously [17,92].

Focus-forming units (FFU) (NP-positive red-colored cells located apart from another one at a distance of two uncolored cells) were then calculated, and the viral titers were expressed as FFU per mL.

### 4.5. Histology

For the histological study, lung specimens were fixed in 10% neutral-buffered formalin (BioVitrum, Moscow, Russia), dehydrated in ascending ethanols and xylols, and embedded in HISTOMIX paraffin (BioVitrum, Moscow, Russia). Paraffin sections (up to 5 µm) were sliced on a Microm HM 355 S microtome (Thermo Fisher Scientific, Waltham, MA, USA) and stained with hematoxylin and eosin. All the images were examined and scanned using an Axiostar Plus microscope equipped with an Axiocam MRc5 digital camera (Zeiss, Oberkochen, Germany) at a magnification of × 200.

The intensity of inflammatory infiltration and metaplasia in the lung tissue was assessed by a semi-quantitative scoring system where 0—no pathological changes, 1—mild inflammation and metaplasia, 2—moderate inflammation and metaplasia, and 3—severe inflammation and metaplasia.

### 4.6. GEO Dataset Analysis

The gene expression profiles associated with influenza A virus (IAV) infection development in mice were acquired from two groups of datasets in the Gene Expression Omnibus [93] database: GSE43302 (3 mock-infected, 5 IAV-infected mice per time point), GSE56433 (3 mock-infected, 4 IAV-infected mice), GSE64798 (4 uninfected, 4 IAV-infected mice), GSE42638 (7 sham, 7 IAV-infected mice), GSE57452 (3 uninfected, 3 IAV-infected mice), and GSE67241 (2 control, 3 IAV-infected mice per time point). Additionally, we have recruited datasets built on the material from human patients with IAV and COVID-19: GSE27131 (IAV, 7 control samples, 7 patients), GSE82050 (IAV, 15 control samples, 24 patients), GSE101702 (IAV, 52 control samples, 44 patients), GSE185576 (IAV, 12 control samples, 56 patients), GSE164805 (COVID-19, 5 control samples, 5 patients), and GSE213313 (COVID-19, 11 control samples, 50 patients). Dataset selection was based on the length of experiment in mouse models (either 7 days post infection (d.p.i.) or 10–12 d.p.i.), or severity (severe or critical) and days after symptom emergence at the date of admission to hospital for the human patient datasets.

The identification of differentially expressed genes (DEGs) between control and IAV/COVID-19-infected samples was performed using GEO2R, a web-service that allows for the conversion of transcriptomic data in the GEO dataset into a list of DEGs through the pipeline in R programming language [94]. A *p*-value < 0.05 and |fold change| > 1.5 were viewed as cutoff values. A Venn diagram analysis of the revealed DEGs was performed using the Bioinformatics and Evolutionary Genomics tool [95].

### 4.7. Gene Association Network Reconstruction and MCODE Analysis

Gene associations were identified based on the data deposited in the Search Tool for the Retrieval of Interacting Genes/Genomes (STRING) database, confidence score > 0.7. Gene association networks were reconstructed based on five sources, including published high-throughput experiments, genomic context prediction, co-expression, automated text mining, and gene associations deposited in other databases. Genes with no association with the main network were removed. Reconstructed gene association networks were visualized as undirected networks with organic layout using Cytoscape. Further analysis of the gene association networks was performed using the MCODE plugin in Cytoscape, using the following parameters: node score cutoff, 0.2; K-core, 2; and max. depth = 100. From each gene association network, only the top interconnected clusters with an MCODE score ≥ 30 were selected for further analysis.

### 4.8. Short Time-Series Expression Miner (STEM) Analysis

The STEM clustering method was used to analyze the expression patterns of DEGs in the lung tissue during the development of IAV infection. Each gene was assigned to the closest profile using a Pearson correlation-based distance metric. A permutation-based test was used to quantify the expected number of genes that would be assigned to each profile to determine the significance level of a given transcriptome profile [20].

### 4.9. Functional Enrichment Analysis

Functional enrichment analysis of the identified DEGs was performed using the ClueGo v.2.5.2 plugin in Cytoscape v.3.10.2 [96]. The DEGs were mapped to the latest versions of Gene Ontology (biological processes), Kyoto Encyclopedia of Genes and Genomes (KEGG), REACTOME, and Wikipathways. The enrichment of functional terms was tested using the two-sided hypergeometric test corrected using the Bonferroni method, followed by selecting significantly enriched terms with a *p*-value < 0.05. To cluster similar functional groups retrieved from different databases in the common pathway-specific modules, GO Term Fusion was used. Functional grouping of the finally selected functional terms was performed using kappa statistics (kappa score ≥ 0.4).

### 4.10. Identification of Potential Transcription Factors Regulating Expression of Core Genes

The potential regulatory transcription factors (TFs) for core genes, regulating the inflammation-to-metaplasia transition in the IAV-infected lung tissue, were identified using the iRegulon plugin in Cytoscape. iRegulon detects the TFs and their targets by scanning known TF-binding promoter motifs as well as the predicted motifs discovered from the Encyclopedia of DNA Elements (ENCODE) Project chromatin immunoprecipitation-sequencing data [97]. The default search options were used.

### 4.11. Quantitative Real-Time PCR (qRT-PCR)

Total RNA was isolated from the lungs of experimental animals using TRIzol Reagent (Ambion, Austin, TX, USA) according to the manufacturer’s recommendation. Briefly, lung tissue was collected in 1.5 mL capped tubes, filled with 1 g of lysing matrix D (MP Biomedicals, Irvine, CA, USA) and 1 mL of TRIzol reagent, and then homogenized using a FastPrep-24TM 5G homogenizer (MP Biomedicals, Irvine, CA, USA), using the QuickPrep 24 adapter. The homogenization was performed at 6.0 m/s for 40 s. After homogenization, the content of the tubes was transferred to new 1.5 mL tubes without lysing matrix. Total RNA extraction was performed according to the TRIzol reagent protocol. First-strand cDNA was synthesized from total RNA in 100 μL of reaction mixture containing 2.5 μg of total RNA, 20 μL of 5× RT buffer (Biolabmix, Novosibirsk, Russia), 250 U M-MuLV-RH revertase (Biolabmix, Novosibirsk, Russia), and 100 μM random hexaprimers (5′-NNNNNN-3′) in a volume of 100 μL. Reverse transcription was performed at 25 °C for 10 min, followed by incubation at 42 °C for 60 min. Finally, reverse transcriptase was terminated at 70 °C for 10 min. Amplification of cDNA was performed in a 25 μL PCR reaction mixture containing 5 μL of cDNA, 12.5 μL of HS-qPCR (2×) master mix (Biolabmix, Novosibirsk, Russia), 0.25 μM each of forward and reverse primers for HPRT and HPRT-specific ROX-labelled probe, 0.25 μM each of forward and reverse gene-specific primers, and FAM-labelled probe (Table 1). Amplification was performed as follows: (1) 94 °C, 2 min; and (2) 94 °C, 10 s; 60 °C, 30 s (50 cycles). The relative level of gene expression was normalized to the level of HPRT according to the ΔΔCt method. Three to five samples from each experimental group were analyzed in triplicate. The relative level of gene expression was determined with a CFX96TM Real-Time system (C1000 TouchTM, Bio Rad laboratories, Hercules, CA, USA).

### 4.12. Analysis of Expression Levels of Core Genes in Cases of COVID-19 with Different Severity

The expression levels of identified core genes in the patients with cases of COVID-19 infection with different severity were identified using bulk RNA-seq in the COVID-19 multi-omics blood ATlas project (COMBATdb). The data in this database include whole blood transcriptomics, plasma proteomics, epigenomics, single-cell multi-omics, immune repertoire sequencing, flow and mass cytometry, and cohort metadata [98]. The results were visualized in COMBATdb (https://db.combat.ox.ac.uk/ (accessed on 10 October 2024)), using the RShiny algorithm.

### 4.13. Survival Analysis of Core Genes

To elucidate the connection between the expression levels of identified core genes and the overall survival of patients with lung adenocarcinoma (LUAD), survival analysis of the identified core genes was carried out using Kaplan–Meier analysis through the OncoLnc tool (http://www.oncolnc.org/ (accessed on 10 October 2024)).

### 4.14. Statistical Analysis

Statistical analysis was performed using the following methods. The correction of the false positive results by the default Benjamin–Hochberg false discovery method during the identification of differentially expressed genes was performed using the GEO2R pipeline. *P*-values during functional enrichment analysis were determined using a two-sided hypergeometric test with Bonferroni step-down corrections in the ClueGo module of Cytoscape software. In the short time-series expression miner (STEM) analysis, the Pearson correlation-based distance metric was used for clustering and the permutation test to determine the significance of each discovered profile. In cases of histological scoring and the qRT-PCR assay, statistical analysis was performed using two-tailed unpaired Student’s t-test in Microsoft Excel 2013 software; *p*-values of less than 0.05 were considered statistically significant. The data are expressed as the mean ± standard error mean.

## 5. Conclusions

The data obtained in this study identified a subset of genes (*Birc5*, *Cdca3*, *Mcm5*, *Nusap1*, *Plk1*, *Prc1*, *Spag5*, *Top2a*, and *Tpx2*) and transcription factors (*E2f1*, *E2f4*, and *Nfya*) constituting the core regulatory structure responsible for the transition from acute inflammation to metaplastic changes in the lung tissue during influenza A virus (IAV) infection. Forming a tightly interconnected gene cluster, the identified genes demonstrate a remarkably similar expression profile throughout the development of IAV infection and are mainly involved in the regulation of the cell cycle, cell proliferation, and apoptosis inhibition, ensuring extremely rapid regeneration and survival of newly formed cells. Considering the association of the identified core genes not only with the development of long-term consequences of IAV infection but also with severe COVID-19 infection and lung adenocarcinoma in patients, the identified genes can be considered as a novel core gene signature associated with pulmonary inflammation-to-metaplasia transition and probable markers of adverse long-term consequences of viral infection in the lung.

## Figures and Tables

**Figure 1 ijms-25-11958-f001:**
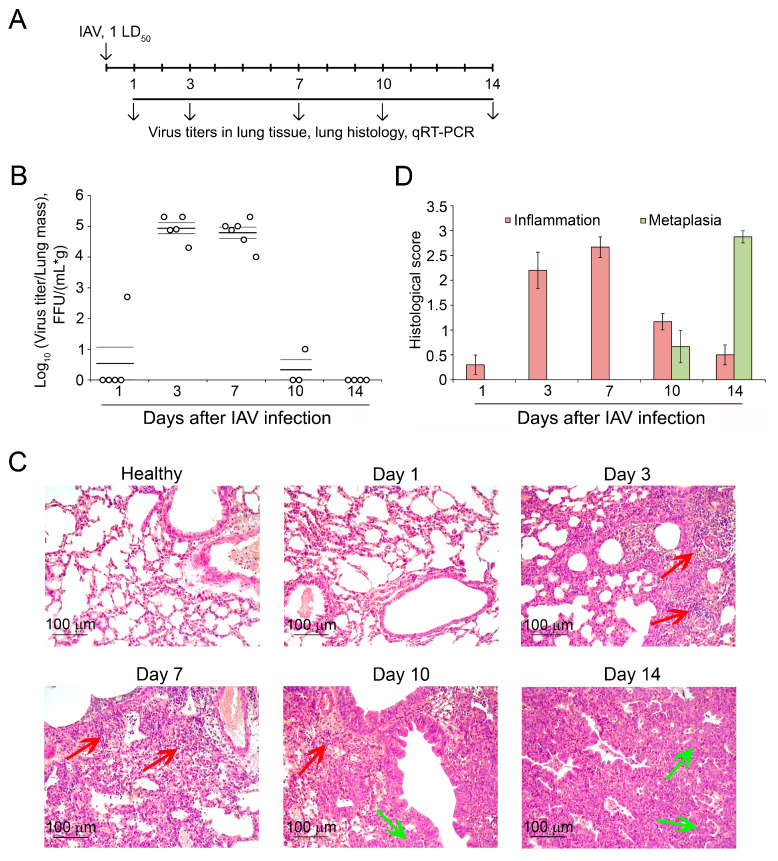
Morphological changes and viral titers in the lungs of mice during development of influenza A virus (IAV) infection. (**A**) Experimental setup. Balb/C mice were intranasally (i.n.) infected with IAV (1 LD_50_). At days 1, 3, 7, 10, and 14 post infection (d.p.i.), the lungs were collected for subsequent analysis. (**B**) Viral titers in the lungs of mice with IAV. (**C**) Representative histological images of IAV-infected lungs on 1, 3, 7, 10, and 14 d.p.i. Hematoxylin and eosin staining, original magnification ×200. Red and green arrows indicate inflammatory infiltration and squamous metaplasia in the lung tissue, respectively. (**D**) Dynamic inflammatory and proliferative changes in the lung tissue of IAV-challenged mice. To assess the intensity of inflammation and squamous metaplasia in the lungs, the semi-quantitative histological scoring system was used: 0—no pathological changes, 1—mild inflammation and metaplasia, 2—moderate inflammation and metaplasia, 3—severe inflammation and metaplasia.

**Figure 2 ijms-25-11958-f002:**
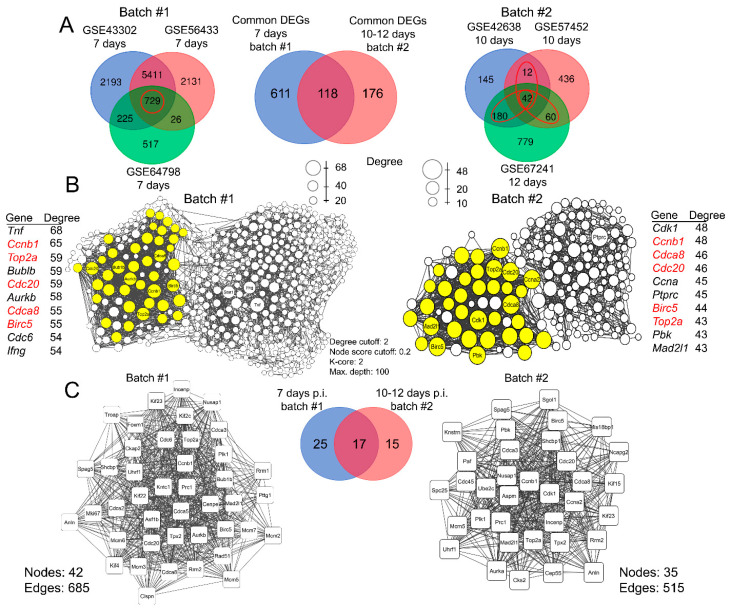
Bioinformatics analysis of key genes involved in the development of influenza virus infection. (**A**) Venn diagram of overlap between differentially expressed genes (DEGs) identified by re-analysis of datasets obtained from murine models of influenza virus infection at 7 (**left**) and at 10–12 (**right**) days after induction, and common DEGs between them (**middle**). DEGs selected for further analysis are circled in red. (**B**) PPI networks reconstructed from DEGs included in the analysis, using STRING database (confidence score ≥ 0.7, maximal number of additional interactors = 0), for batch #1 (**left**) and batch #2 (**right**). The sizes of nodes indicate roughly the centrality (degree) of each node. Additionally, clusters of the most interconnected DEGs were identified using MCODE plugin in Cytoscape software v.3.10.2. Yellow color indicates the nodes in the top-score MCODE cluster. Red gene titles indicate genes common for both gene association networks. (**C**) Detailed PPI networks, reconstructed from the DEGs in the MCODE clusters, for batch #1 (**left**) and batch #2 (**right**), and Venn diagram showing common DEGs between MCODE clusters (**middle**).

**Figure 3 ijms-25-11958-f003:**
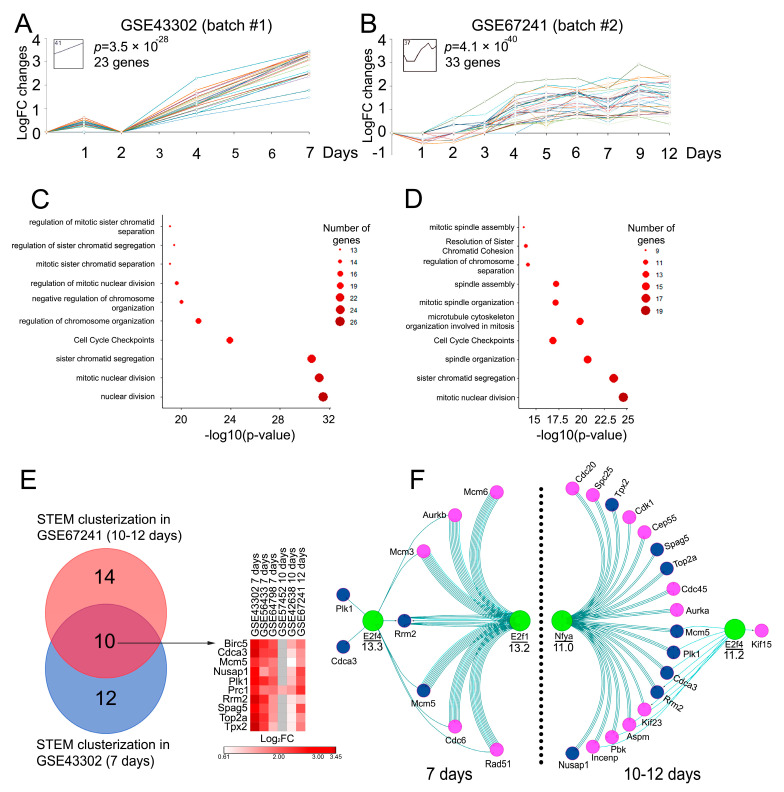
Bioinformatics analysis of key genes and corresponding transcription factors involved in the development of influenza virus infection. (**A**,**B**) Graphs showing the dynamics of DEG expression levels during the course of influenza infection in the most representative GSE dataset, obtained through the STEM clusterization analysis. Only DEGs in the same profile as indicated by STEM are shown on the graphs. Different graph colors indicate different genes. (**C**,**D**) Functional analysis of the DEGs in the STEM clusterization profiles. Enrichment for Gene Ontology (biological processes), KEGG, REACTOME, and WikiPathways terms were performed using ClueGO plugin in Cytoscape. Only pathways with *p* < 0.05 after Bonferroni step-down correction for multiple testing were included. (**E**) Venn diagram demonstrating common genes between DEGs identified at 10–12 d.p.i. (upper Venn diagram) and 7 d.p.i. (lower Venn diagram), and heatmap showing expression levels of identified DEGs in all analyzed datasets. (**F**) Identification of transcription factors regulating the expression of identified DEGs through iRegulon plugin in Cytoscape software. Only transcription factors and DEGs present in motifs with NES higher than 10 are shown. Green nodes—transcription factors, blue nodes—DEGs identified in (**E**), pink—DEGs absent in performed Venn diagram analysis. Numbers under lines near transcription factors indicate NES scores.

**Figure 4 ijms-25-11958-f004:**
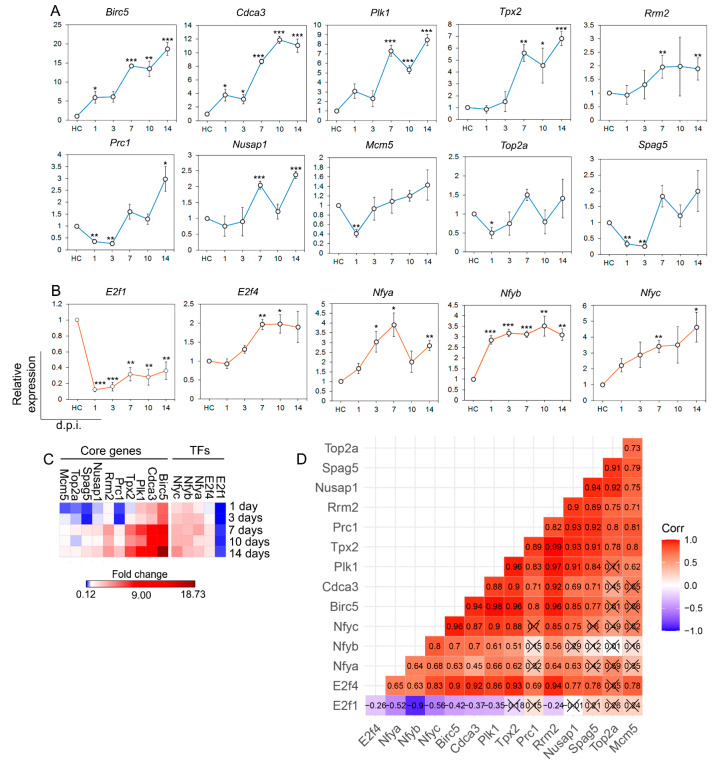
Analysis of expression profiles of genes and transcription factors identified by bioinformatics analysis in the lung tissue of mice challenged with IAV. (**A**,**B**) The graphs show qRT-PCR data for healthy lungs (healthy control, HC) and influenza-challenged lungs at several time points after infection. Blue graph lines are DEGs, orange graph lines are TFs. Expression levels were normalized to the expression level of hypoxanthine phosphoribosyltransferase (HPRT) used as a reference gene. Three to five samples from each group were analyzed in triplicate. The data are shown as mean ± standard error mean, *—*p* < 0.05, **—*p* < 0.01, ***—*p* < 0.001. (**C**) Heatmap visualizing expression level changes during influenza virus infection development. The heatmap was constructed using Morpheus web-application (https://software.broadinstitute.org/morpheus/ (accessed on 17 May 2024)). (**D**) Correlation matrix visualizing correlations between expression profiles of identified DEGs and TFs throughout IAV infection development. The correlation coefficient was calculated according to the Pearson formula through ggcorrplot R package, and the matrix was visualized through ggplot2 R package. Cells with non-significant correlation coefficients are crossed out.

**Figure 5 ijms-25-11958-f005:**
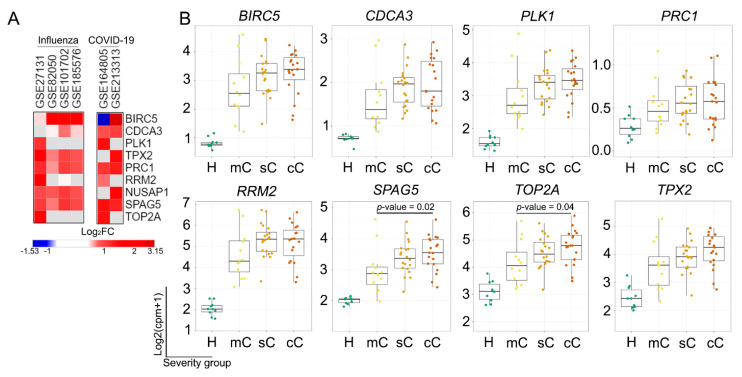
Expression profiles of genes identified in the lung tissue of IAV-infected mice and in the peripheral blood of influenza and COVID-19 human patients. (**A**) Heatmap showing expression of identified DEGs in public datasets, obtained from human patients with IAV and COVID-19 infections. Heatmap was constructed using Morpheus web-tool of the Broad Institute (https://software.broadinstitute.org/morpheus/ (accessed on 10 August 2024)). (**B**) Expression level of DEGs of interest in peripheral blood of COVID-19 patients according to bulk-RNA seq analysis. Graph was constructed using COMBATdb comparing blood from healthy volunteers (H) with blood from mild (mC), severe (sC), and critical (cC) COVID-19 patients. Sample inclusion strategy was one priority sample at maximum severity per individual. FDR ≤ 0.05.

**Figure 6 ijms-25-11958-f006:**
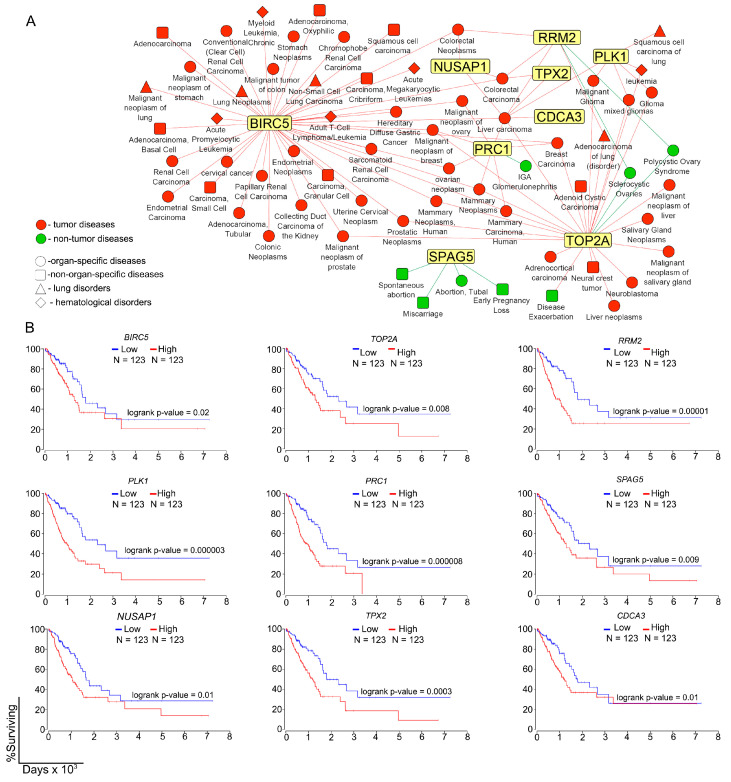
Association between identified DEGs of interest and human neoplasms. (**A**) Network, including connections between identified DEGs and human diseases. Disease nodes include neoplasm (red) and non-neoplasm diseases (green) with additional classification based on organ specificity: organ-specific (circle), non-organ-specific (square), lung disorders (triangle) and hematological disorders (rhombus). Search was performed in DisGeNET curated database, all sources included. (**B**) Correlation between high and low expression of identified DEGs of interest and survival rate of patients with lung adenocarcinoma (LUAD). Comparison was performed between patients in 25th percentile of high and low expression levels using TCGA database and OncoLnc web-server [91].

**Table 1 ijms-25-11958-t001:** Sequences of the primers used in this study.

Gene	Type	Sequence
*Birc5*	Forward	5′-AAGGAATTGGAAGGCTGGG-3′
	Probe	5′-((5,6)-FAM)-ACCCGATGACAACCCGATAGAGGA-BHQ1-3′
	Reverse	5′-TTCTTGACAGTGAGGAAGGC-3′
*Cdca3*	Forward	5′-GGACCTTGACACTACGACAG-3′
	Probe	5′-((5,6)-FAM)-ACCAGGACCACGACAAGGAAAATCAG-BHQ1-3′
	Reverse	5′-AGGAATCAGAACAGGGCTTTC-3′
*Mcm5*	Forward	5′-TCATCTCCAAGAGCATTTCCC-3′
	Probe	5′-((5,6)-FAM)-CAGGGTCTCCCAACATCAGCAAGT–BHQ1-3′
	Reverse	5′-GCACTTCTCCACAAACTTCAG-3′
*Nusap1*	Forward	5′-CCGCTCTGTTTTGAGTTGTG-3′
	Probe	5′-((5,6)-FAM)-CGTTCTGCAATCTCGGTGATTCCCA–BHQ1-3′
	Reverse	5′-GATTCAGGTGTGCTTTCAAGG-3′
*Plk1*	Forward	5′-CTTCACTTCTGGCTACATCCC-3′
	Probe	5′-((5,6)-FAM)-CCCGTCTCCCTATTACCTGCCTCA–BHQ1-3′
	Reverse	5′-ATGGCCTCATTTGTCTCCC-3′
*Prc1*	Forward	5′-CTCGATTCTGGACAGCTTGG-3′
	Probe	5′-((5,6)-FAM)-CCTCCGGGAAATATGGGAACTAATTGGG–BHQ1-3′
	Reverse	5′-CTCGGTTCTTTGTAGCCTCTG-3′
*Rrm2*	Forward	5′-TGGCTGACAAGGAGAACAC-3′
	Probe	5′-((5,6)-FAM)-TAACGGCTCATCCTCAACGCTGG–BHQ1-3′
	Reverse	5′-GATGGGAAAGACAACGAAGC-3′
*Spag5*	Forward	5′-GGACAGCAAGGAGATTAGACAG-3′
	Probe	5′-((5,6)-FAM)-TTAGGGACAGGTGAAGCAAGGACAAC-BHQ1-3′
	Reverse	5′-CAGCACACGAGAACAGGAG-3′
*Top2a*	Forward	5′-TGGCTTCTAGGAATGCTTGG-3′
	Probe	5′-((5,6))-FAM)-TCTGACCCTGTGAAAGCCTGGAAAG-BHQ1-3′
	Reverse	5′-TCTTCATCTGGAACCTTCTGC-3′
*Tpx2*	Forward	5′-GAGATAGAGAAAAGGCTGCGG-3′
	Probe	5′-((5,6)-FAM)-CTGTCTTGGTAACCTGGCTCGTGG-BHQ1-3′
	Reverse	5′-CTTGGGATGTTGCTTGATTCG-3′
*E2f1*	Forward	5′-GAAACGGAGGCTGGATCTG-3′
	Probe	5′-((5,6)-FAM)-CCGGAGATTTCACACCTTTCCCTGG–BHQ1-3′
	Reverse	5′-AGTGAGGTTTCATAGCGTGAC-3′
*E2f4*	Forward	5′-ATCAAAGCAGACCCCACAG-3′
	Probe	5′-((5,6)-FAM)-TCCCGGAGACCACGATTACATCTACAA–BHQ1-3′
	Reverse	5′-CACAGACACCTTCACTCTCG-5′
*Nfya*	Forward	5′-GAAATACCTCCATGAGTCTCGG-3′
	Probe	5′-((5,6)-FAM)-CAGCTTGGTTTGGATCCTGCATGTG-BHQ1-3′
	Reverse	5′-TCTGTGTCATGGCTTCTTCG-3′
*Nfyb*	Forward	5′-TGCCATACCTCAAACAGGAAAG-3′
	Probe	5′-((5,6)-FAM)-TCACACATTCCTGAACACATTCTTTGGC-BHQ1-3′
	Reverse	5′-ATTGTCTTCCGCTTCTCCTG-5′
*Nfyc*	Forward	5′-AGCAGATCATTACCAACACAGG-3′
	Probe	5′-((5,6)-FAM)-TTGGACAACTTGGGTGCCTGATACA-BHQ1-3′
	Reverse	5′-GGTAGCAAGTGTCTGGATCTG-3′
*HPRT*	Forward	5′-CCCCAAAATGGTTAAGGTTGC-3′
	Probe	5′-((5,6)-ROX)-CTTGCTGGTGAAAAGGACCTCTCGAA-BHQ2-3′
	Reverse	5′-AACAAAGTCTGGCCTGTATCC-3′

## Data Availability

All publicly available cDNA microarray data can be obtained from Gene Expression Omnibus by querying the accession number of interest.

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
