# Peer review of "Novel Core Gene Signature Associated with Inflammation-to-Metaplasia Transition in Influenza A Virus-Infected Lungs"

_ijms, 2024, doi:10.3390/ijms252211958_

Round 1
Reviewer 1 Report
Comments and Suggestions for Authors
The authors discovered through morphological changes in the lung tissue of mice that within a 7-10 day window after IAV infection, the lung tissue undergoes a transformation from inflammation to metaplasia. By analyzing six datasets from the GEO database, they identified five key genes involved in the development of influenza virus infection. Then, they selected representative datasets at 7 days and 12 days post-infection and used the short time expression miner tool to cluster gene expression in a time-dependent manner, identifying 10 core genes. Subsequently, they identified potential transcription factors regulating the expression of core differentially expressed genes (DEGs) involved in the transition from inflammation to metaplasia during IAV infection development. They validated and analyzed these core genes and transcription factors using qPCR. These findings provide new insights into the transition from lung inflammation to metaplasia after IAV infection and are logically rigorous as a whole.
Minor problem:
The qPCR results are the most important data, therefore, the primer sequences need to be listed.
Author Response
Dear Reviewer 1,
Thank you for the valuable comment that helped us to improve the manuscript.
Comment 1:
The qPCR results are the most important data, therefore, the primer sequences need to be listed.
Response 1:
We include the primer sequences used in the study to the Materials and methods section (please, see Table 1, line 705, revised parts are marked by yellow).
Reviewer 2 Report
Comments and Suggestions for Authors
The manuscript titled “Novel core gene signature associated with inflammation-to-metaplasia transition in influenza A virus-infected 3 lungs” identifies a set of core genes that are found to be differentially regulated during transition from inflamed state to metaplastic state. In this study, they identify core gene signature not only with the IAV-induced inflammation-to-metaplasia transition but also with other pulmonary epithelial infections such as Covid-19. Overall, they make several findings using a variety of computational tools that underscore the possible involvement of genes like SPAG5 and TOP2A, that are implicated in a range of cellular processes. They even corroborate the findings using human clinical sample sets, adding credibility to the findings. This manuscript highlights the possible involvement of several new genes in the transition of with inflammation-to-metaplasia during virus infections using transcriptomics and discussing how these genes might be involved in the transition will significantly improve the quality of the manuscript. Most importantly, elaborating on how nuclear factors such as TOP2A during infection by predominantly RNA viruses such as SARS-CoV-2, will help improve the credibility of the findings. Overall, I recommend that thus manuscript be accepted for publication after minor revisions in the discussion section, elaborating on the functional relevance of at least SPAG5 and TOP2A, as the whole argument presented is based on transcriptomic data and still needs to be corroborated by ELISA.
Author Response
Dear Reviewer 2,
Thank you for the valuable comments and suggestions that helped us to improve the manuscript. We carefully take into account all mentioned issues, revised parts are marked by yellow.
Comment 1:
Most importantly, elaborating on how nuclear factors such as TOP2A during infection by predominantly RNA viruses such as SARS-CoV-2, will help improve the credibility of the findings.
Response 1:
We are deeply grateful for this comment. Indeed, interaction of nuclear factors such as TOP2A with RNA viruses including Influenza and SARS-CoV-2 is complex process with many signaling pathways involved. We discussed literature data concerning role of SPAG5 in cytoprotective response in COVID-19, as well as its possible involvement in the regulation of COVID-19-associated cytokine storm and subsequent lung injury (please, see lines 518-524). In the case of TOP2A, we demonstrated reported involvement of TOP2A in RNA virus infection via controlling the replication and transcription of viral RNA genome (please, see lines 524-530).
Comment 2:
Overall, I recommend that thus manuscript be accepted for publication after minor revisions in the discussion section, elaborating on the functional relevance of at least SPAG5 and TOP2A, as the whole argument presented is based on transcriptomic data and still needs to be corroborated by ELISA.
Response 2:
We fully agree that basing wide-reaching conclusions on transcriptomic data alone is ill-advised, so we tried to justify functional relevance of aforementioned genes in proliferation, cell division and metaplasia. Unfortunately, we were not able to perform ELISA analysis due to difficulties in obtaining relevant kits, so we performed literature analysis demonstrating involvement of SPAG5 and TOP2A in cell proliferation and survival corroborated by western blotting and flow cytometry. Please, see lines 434-440 for SPAG5, lines 441-444 for TOP2A and lines 449-453.
Additionally, we included published data concerning involvement of the most susceptible to IAV infection genes, namely Birc5, Cdca3, Tpx2 and Plk1, in the development of inflammation-to-metaplasia transition, hyperproliferation and metaplastic conditions in patients (please, see lines 398-407).